# Interpretation of NO₃-N₂O₅ observation via steady state in high aerosol air mass: The impact of equilibrium coefficient in ambient conditions

Xiaorui Chen[1], Haichao Wang[3,4], Keding Lu[1,2]

[1]State Key Joint Laboratory of Environmental Simulation and Pollution Control, College of Environmental Sciences and Engineering, Peking University, Beijing, China.

[2]The State Environmental Protection Key Laboratory of Atmospheric Ozone Pollution Control, College of Environmental Sciences and Engineering, Peking University, Beijing, China

[3]School of Atmospheric Sciences, Sun Yat-sen University, Zhuhai, 519082, China

[4]Guangdong Provincial Observation and Research Station for Climate Environment and Air Quality Change in the Pearl River Estuary, Key Laboratory of Tropical Atmosphere-Ocean System, Ministry of Education, Southern Marine Science and Engineering Guangdong Laboratory (Zhuhai), Zhuhai, 519082, China

*Correspondence to:* Haichao Wang (wanghch27@mail.sysu.edu.cn), Keding Lu (k.lu@pku.edu.cn)

**Abstract.** Steady state approximation for interpreting NO₃ and N₂O₅ has large uncertainty under complicated ambient conditions and could even produces incorrect results unconsciously. To provide an assessment and solution to the dilemma, we formulate data sets based on in-situ observations to reassess the applicability of the method. In most of steady state cases, we find a prominent discrepancy between $K$eq (equilibrium coefficient for reversible reactions of NO₃ and N₂O₅) and correspondingly simulated $[N_2O_5]/([NO_2]\times[NO_3])$, especially under high aerosol conditions in winter. This gap reveals the accuracy of $K$eq has a critical impact on the steady state analysis in polluted region. In addition, the accuracy of $\gamma(N_2O_5)$ derived by steady state fit depends closely on the reactivity of NO₃ ($k$NO₃) and N₂O₅ ($k$N₂O₅). Based on a complete set of simulations, air mass of $k$NO₃ less than 0.01 s⁻¹ with high aerosol and temperature higher than 10°C is suggested to be the best suited for steady state analysis of NO₃-N₂O₅ chemistry. Instead of confirming the validity of steady state by numerical modeling for every case, this work directly provides appropriate concentration ranges for accurate steady state approximation, with implications for choosing suited methods to interpret nighttime chemistry in high aerosol air mass.

## 1 Introduction

Nitrate radical ($NO_3$), an extremely reactive species prone to build up at night, is an ideal candidate for steady state analysis in combine with dinitrogen pentoxide ($N_2O_5$) due to fast equilibrium reactions between them (R1).

$$NO_2 + NO_3 + M \; \rightarrow \; N_2O_5 + M \qquad\qquad\qquad\qquad\qquad (R1a)$$

$$N_2O_5 + M \; \rightarrow \; NO_2 + NO_3 + M \qquad\qquad\qquad\qquad\qquad (R1b)$$

Under the steady state condition, the lifetime of $NO_3$ (denoted as $\tau_{ss}(NO_3)$) can be calculated as the ratio of $NO_3$ concentration over the production rate ($k_{NO_2+O_3}[NO_2][O_3]$) or over the removal rate of both $NO_3$ and $N_2O_5$, as indicated in Eq. (1). A similar representation of $N_2O_5$ steady state lifetime is also shown in Eq. (2). The loss frequencies of various sink pathways of $NO_3$ and $N_2O_5$ are integrated as total first-order in the following equations, represented by $kNO_3$ and $kN_2O_5$ term. Briefly, the $kNO_3$ is contributed by the reaction of $NO_3$ radical with NO and hydrocarbons and uptake on particles at night, ranging from hundredths of $s^{-1}$ to several $s^{-1}$ depending on the air mass. Due to its large rate constant with NO, the concentration usually dominates the lifetime of $NO_3$ radical in urban areas with fresh NO emission. Otherwise, the reactions with hydrocarbons, especially unsaturated hydrocarbons, is preferential for $NO_3$ in rural areas. The $K$eq denotes the equilibrium coefficient for reactions R1a and R1b, used to be derived by Eq. (3).

$$\tau_{ss}(NO_3) \equiv \frac{[NO_3]}{k_{R1}[NO_2][O_3]} \approx (k_{NO_3} + K_{eq}[NO_2]k_{N_2O_5})^{-1}, \qquad\qquad (1)$$

$$\tau_{ss}(N_2O_5) \equiv \frac{[N_2O_5]}{k_{R1}[NO_2][O_3]} \approx (k_{N_2O_5} + \frac{k_{NO_3}}{K_{eq}[NO_2]})^{-1}, \qquad\qquad (2)$$

$$K_{eq} = \frac{k_{R1a}}{k_{R1b}} = \frac{[N_2O_5]}{[NO_2][NO_3]}, \qquad\qquad\qquad\qquad\qquad (3)$$

Numerous works have taken the advantage of the steady state calculation to quantify the total first-order loss rate for $NO_3$ or $N_2O_5$ such that they drew conclusions about the oxidation capacity and reactive nitrogen budgets contributed by this chemical system (Allan et al., 1999;Allan et al., 2000;Carslaw et al., 1997;Platt et al., 1984;Vrekoussis et al., 2007;Wang et al., 2013). Since the steady state approximation was used to interpret atmospheric observation of $NO_3$-$N_2O_5$ (Brown, 2003; Platt et al., 1981), this method was also widely implemented to quantify $N_2O_5$ uptake coefficient ($\gamma(N_2O_5)$) (Brown et al., 2009;Brown et al., 2003;Li et al., 2020;McDuffie et al., 2019;Phillips et al., 2016;Wang et al., 2017a;Wang et al., 2017c;Wang et al., 2020a).

However, with the influence induced by complicated atmospheric conditions and emission, the steady state in ambient air mass will not always be the case (as illustrated in Text

S1 and Figure S1). These situations are prevalent in nocturnal boundary layer (Phillips et al., 2016;Stutz et al., 2004;Wang et al., 2017a;Wang et al., 2017c) and therefore increase the difficulty of applying steady state directly on $NO_3$-$N_2O_5$ observation data, whereas few studies have systematically characterized the error source and application conditions of this method (Brown et al., 2009).

Due to faster approach to equilibrium than steady state, the application of $K$eq in calculation steady state equations seems to be reasonable (Brown et al., 2003). For example, the ambient $NO_3$ concentration was usually calculated based on ambient $N_2O_5$ concentration with $K$eq×[$NO_2$] when determining their budgets or characterizing the lifetime or sink attribution of these two reactive nitrogen compounds (Brown et al., 2011;Osthoff et al., 2006;Wang et al., 2018;Wang et al., 2017c;Wang et al., 2017d;Yan et al., 2019). In addition, the mathematical conversion between $NO_3$ and $N_2O_5$ concentration via $K$eq coefficient can simplified the calculation in the iterative box model, which derives $\gamma(N_2O_5)$ by iterating its value in the model until the predicted $N_2O_5$ concentration matches the observation (Wagner et al., 2013;Wang et al., 2020b). However, considerable uncertainty could be associated with the quantification of $K$eq and its different parameterizations (Cantrell et al., 1988;Pritchard, 1994). The impact of $K$eq value on steady state fit or concentration conversion have not been explored to date in the analysis of $NO_3$-$N_2O_5$ steady state.

In this study, we formulate a half artificial dataset with expected properties based on field campaigns. Specifically, most of species contained in the dataset are observed values while only $NO_3$ and $N_2O_5$ were calculated by the steady state model (illustrated in the section 2.2). With the dataset, we illustrate the reasons for deviation of parameterized $K$eq from $[N_2O_5]/([NO_2]\times[NO_3])$ in ambient conditions, the possible uncertainties of linear fit based on steady state equations Eq. (4) and Eq. (5) (the related variables are explained in section 2.1) resulted from different $K$eq, and the influence of relevant atmospheric variables on $\gamma(N_2O_5)$ derivation via steady state method. Furthermore, a series of ambient condition tests specify the exact ranges suited for steady state analysis according to not only the validity of steady state but also $K$eq values, which optimizes the validity check by numerical modeling in previous research (Brown et al., 2009;Brown et al., 2003) and develops complete standard for data filtering.

## 2 Methods

### 2.1 γ(N₂O₅) derivation by steady state approximation

The framework of steady state approximation for NO₃-N₂O₅ system is basically built on its chemical production and removal pathways, in case of extremely weak physical processes (e.g. transport, dilution and deposition) relative to its chemical processes. With simultaneous measurements of NO₃, N₂O₅ and relevant precursor concentrations, the steady state lifetime $\tau_{ss}(NO_3)$ and $\tau_{ss}(N_2O_5)$ can be quantified for a targeted period as shown in Eq. (1) and Eq. (2). By substituting the $k_{N_2O_5}$ with $0.25 \times c \times S_a \times \gamma(N_2O_5)$, the $\gamma(N_2O_5)$ and the reactivity of NO₃ ($k_{NO_3}$, including the reactions of NO₃ with NO and hydrocarbons) can therefore be determined by Eq. (4) and Eq. (5).

$$\tau_{ss}^{-1}(NO_3) \approx k_{NO_3} + 0.25cS_aK_{eq}[NO_2]\gamma(N_2O_5), \tag{4}$$

$$\left(0.25cS_a\tau_{ss}(N_2O_5)\right)^{-1} \approx \gamma(N_2O_5) + k_{NO_3}\left(0.25cS_aK_{eq}[NO_2]\right)^{-1}, \tag{5}$$

Here $c$ represents the mean molecular velocity of N₂O₅, Sa represents the aerosol surface area and the $K_{eq}$ is calculated from the rate constant of reversible reactions R1a ($k_{R1a}$) and R1b ($k_{R1b}$), which is a temperature-dependent parameter. It should be noted that the photolysis of NO₃ is not considered in the $k_{NO_3}$ due to weak radiation at night and the homogeneous hydrolysis was also ignored due to its small contribution in comparison to heterogeneous pathway, similar presumption was also implemented in previous studies (Brown et al., 2009;Mentel et al., 1996;Wahner et al., 1998). In the form of these two equations, the potential covariance between Sa and NO₂ concentration can be avoided to decrease the uncertainty (Brown et al., 2009). By fit to these two equations, γ(N₂O₅) can be directly derived from slope of the plot of $\tau_{ss}^{-1}(NO_3)$ against $0.25cS_aK_{eq}[NO_2]$ or from intercept of the plot of $\left(0.25cS_a\tau_{ss}(N_2O_5)\right)^{-1}$ against $\left(0.25cS_aK_{eq}[NO_2]\right)^{-1}$ respectively. In the following analysis, the linear fit based on Eq. (5) is preferred in steady state approximation.

### 2.2 Steady state model and half-artificial datasets

The steady state model is reformed from 0-dimension box model to produce NO₃ and N₂O₅ which are in steady state as far as possible. It is constrained by measurements of NO, NO₂, O₃, CO, CH₄, VOCs, HCHO, Sa, relative humidity (RH), temperature (T), pressure, coupled with Regional Atmospheric Chemistry Mechanism, version 2 (RACM2). Each data point is treated as an independent air mass, aging 10 hours and keeping input constraint unchanged. As NO₃-N₂O₅ chemistry, the interest of this work, usually shows marked impacts during the night, only the time periods with negligible photolysis frequency are under consideration. In

the standard simulation (herein referred as Mod0), the uptake coefficient of $N_2O_5$ is set to 0.02,
as a reasonable value of literatures (Brown et al., 2006;Chen et al., 2020;McDuffie et al.,
2018;Morgan et al., 2015;Phillips et al., 2016;Wagner et al., 2013;Wang et al., 2017c;Yu et al.,

127      2020).

128         Two half-artificial datasets are derived from PKU2017 and TZ2018 field campaigns (see

Text S2) based on steady state model for analysis in the following sections. The simulated $NO_3$
and $N_2O_5$ and other observed values used for the constraints of steady state model jointly
formulate these half-artificial datasets. Specifically, the $NO_3$ and $N_2O_5$ concentration in this
dataset are the output of the steady state model simulation, and guaranteed to be in steady state
with respect to other observed precursors. To verify the steady state of $NO_3$ and $N_2O_5$ for each
data point, we filtered the data set according to deviation between steady state lifetime of $N_2O_5$
( $\tau_{ss}(N_2O_5) = \frac{[N_2O_5]}{k_{R1}[NO_2][O_3]}$ ) and calculated lifetime of $N_2O_5$ ( $\tau_{calc}(N_2O_5) = (k_{N_2O_5} +$
$\frac{k_{NO_3}}{K_{eq}[NO_2]})^{-1}$). If the deviation exceeds 10% for a data point, it will be excluded from the
following analysis. We presume that if any data point outputted from the model is still out of
steady state in terms of $NO_3$ and $N_2O_5$, the sink rate constant of air mass represented by this
data point should be too weak for steady state analysis within a reasonable timescale. In
addition, the data higher than 5 ppbv NO is filtered out in the following calculation, since the
resulting large variation of $kNO_3$ can bias the linear fit even though the $NO_3$ and $N_2O_5$
approach the steady state rapidly under high NO (discussed in 3.2). The fraction of excluded
data is less than 8%, which are expected to have little influence on our results. The calculated
nighttime loss fraction accounted by $NO_3$ and $N_2O_5$ show large discrepancy (see Text. S3 and
Figure S2) between these two half-artificial datasets, which provide us a good opportunity to
investigate the impacting factors on steady state approximation across different conditions.

147         Rather than using observation data directly, a half-artificial dataset can provide larger

amount of valid data for steady state analysis with known $\gamma(N_2O_5)$ value. Besides, this method
avoids the impacts from steady state deviation, which helps to analyze the factors influencing
$\gamma(N_2O_5)$ quantification via steady state approximation backwards from a known steady state
condition.
**3 Results and discussion**
**3.1 Varying equilibrium coefficient under steady state**
The rates of $NO_3$-$N_2O_5$ reversible reactions are expected to be equal for the steady state case,
so that the equilibrium coefficient $K$eq can be determined from either the rate constant ratio of
R1a and R1b or the ratio of $[N_2O_5]/([NO_2] \times [NO_3])$. Although this approach is reasonable
under ideal conditions, the exactly same rates between reversible reactions and the following
calculation based on $K$eq scaling are not so appropriate for ambient atmosphere where the
removal pathway for $NO_3$-$N_2O_5$ are not negligible, especially under the high aerosol loading
condition. The $NO_3$-$N_2O_5$ achieves steady state after 1.5-hours evolution, when concentration
and rates remain constant (Figure 1). In this simulation, the starting mixing ratios of $NO_2$ and
$O_3$ are 10 and 23 ppbv respectively, which is the average level for the nighttime conditions in
PKU2017. The concentration of these two precursors are held constant in the simulation to
better illustrate the influence of removal rates. This result will stay almost the same no matter
these starting values are initialized to be constant or allowed to vary. Under steady state, the
net equilibrium reaction rate in Figure 1(b)&(c) stays negative and positive for $NO_3$ and $N_2O_5$
respectively. Besides, the absolute values and difference of the forward and backward reaction
rates remain unchanged after achieving steady state. This result is similar with a previous
numerical calculation study (Brown et al., 2003), while the deviation between reversible
reaction rates becomes larger in our case.

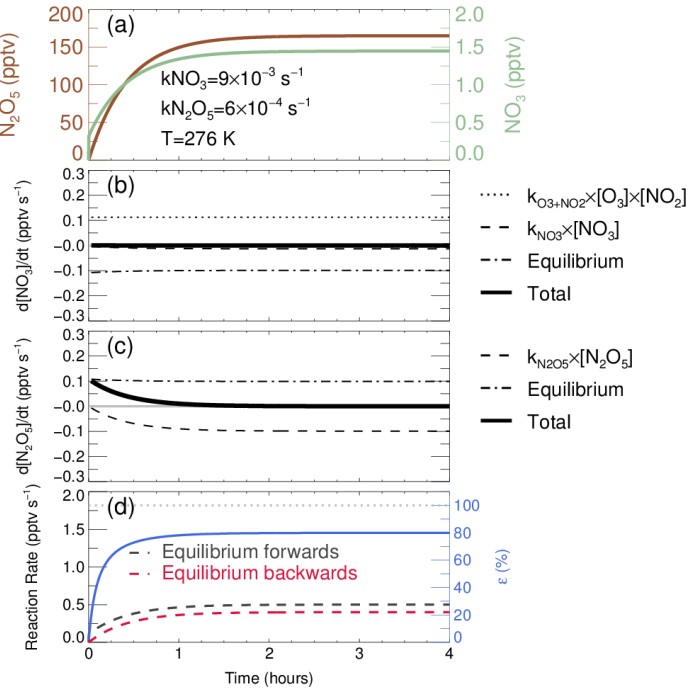


**Figure 1.** Evolution of $NO_3$-$N_2O_5$ system simulated by steady state model for an average case. (a) Temporal
profiles of $N_2O_5$ and $NO_3$, the constraint of simulation is displayed as the text; (b) Evolution of d[$NO_3$]/dt
calculated from source of $k_{O3+NO2} \times [O_3] \times [NO_2]$, sink of $kNO_3 \times [NO_3]$ and equilibrium terms, detailed in
the text; (c) Evolution of d[$N_2O_5$]/dt calculated from equilibrium terms, sink of $kN_2O_5 \times [N_2O_5]$; (d) Forward
($N_2O_5$ formation) and backward ($N_2O_5$ decomposition) equilibrium rate are represented as black and red
dash lines, the equilibrium completeness $\varepsilon$ is calculated by the ratio of backward rate over forward rate,
shown as blue full line.

179       In this case, the original equilibrium is imperfect realized (a perfect realization of the
original equilibrium condition is that $K$eq and the ratio of $[N_2O_5]/([NO_2] \times [NO_3])$ are
equivalent as Eq. (6)), leading to errors on projection of $NO_3$ and $N_2O_5$ concentration via $K$eq
$\times [NO_2]$. In fact, we note that a new equilibrium between $NO_3$ and $N_2O_5$ is developed with
constant but unequal rates. Under this new equilibrium condition, the ratio of R1b reaction
rate (the red dash line in Figure1(d)) over R1a reaction rate (the black dash line in Figure1(d))
can be regarded as the degree of approaching original equilibrium (the blue line in Figure1(d)).
In addition, this value is also the ratio of $[N_2O_5]/([NO_2] \times [NO_3])$ against original $K$eq,
therefore we defined this ratio as a correction factor $\varepsilon$, implemented to calculate accurate
$[N_2O_5]/([NO_2] \times [NO_3])$ with significant $N_2O_5$ removal pathways. The value of $K$eq after
scaled by $\varepsilon$ can be used for converting the concentration of $NO_3$ and $N_2O_5$ via Eq. (6):
$$\varepsilon \times K_{eq} = \varepsilon \times \frac{k_{R1a}}{k_{R1b}} = \frac{[N_2O_5]}{[NO_2][NO_3]}, \qquad\qquad\qquad\qquad\qquad (6)$$

191       Sensitivity tests are conducted to demonstrate the dependence of $\varepsilon$ on relevant variables
based on steady state model. The average ambient conditions observed at wintertime PKU site
and summertime TZ site are taken as basic constraint for sensitivity tests (Table S2),
respectively. By separately altering variables, such as $NO_2$, $O_3$, $k$N$_2$O$_5$, $k$NO$_3$ and T, the
sensitivity of $\varepsilon$ value can be obtained as shown in Figure 2 and Figure S4. The $\varepsilon$ value
depends primarily on $k$N$_2$O$_5$ and $T$ in both scenarios, where $\varepsilon$ increases with $T$ (approaching
1 under relatively high $T$) and decreases with $k$N$_2$O$_5$. In comparison, the $\varepsilon$ value behaves
insensitive to $k$NO$_3$ as well as $NO_2$ and $O_3$ concentration, at least within the range of reasonable
ambient conditions. High $k$N$_2$O$_5$ is resulted from high aerosol events, usually occur in winter
accompanied with low temperature and high relative humidity in some populated areas
(Baasandorj et al., 2017;Huang et al., 2014;Wang et al., 2017b;Wang et al., 2014), further
decreasing the accuracy of original $K$eq values. It can be inferred that in order to accurately
interpreting relationship of $NO_3$ and $N_2O_5$, calculation relying on equilibrium equation and
steady state approximation should consider the dependence of $\varepsilon$ on ambient conditions.

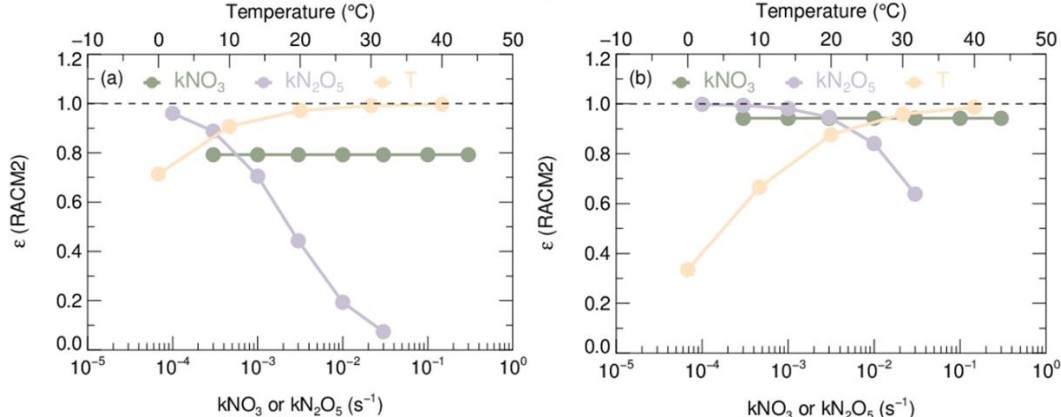


**Figure 2.** Sensitivity plot of $k\mathrm{NO_3}$, $k\mathrm{N_2O_5}$ and Temperature (T) against coefficient $\varepsilon$. The trace of T is plotted against the upper horizontal axis and the traces of the other two parameters are plotted against the lower horizontal axis. (a) Basic model condition is according to typical winter condition of PKU2017; (b) Basic model condition is according to typical summer condition of TZ2018. Basic model conditions including $k\mathrm{NO_3}$, $k\mathrm{N_2O_5}$ and Temperature (T) are shown in Table S2. It should be noted that the provided ranges of each factor do not exactly equal to but encompass the ambient conditions encountered during the two campaigns.

Even if $K$eq value serves as a good representation of the ratio of $[\mathrm{N_2O_5}]/([\mathrm{NO_2}]\times[\mathrm{NO_3}])$ or $\varepsilon$ can be readily quantified on field, the discrepancy among different database in calculating $K$eq still increase the uncertainties of $\mathrm{NO_3}$-$\mathrm{N_2O_5}$ calculation through steady state approximation or equilibrium, which has not been carefully considered. Here, we apply a set of uniform formulas to describing $k_{\mathrm{R1a}}$ and $k_{\mathrm{R1b}}$ (see Text. S4) from preferred values of several popular atmospheric chemistry mechanisms (Mozart, CB05, Saprc07, RACM2 and kinetic databases JPL2015 as well as IUPAC2017) and finally calculating $K$eq. As is shown in Figure S5 and Figure S6, $K$eq variations derived from these six different databases reflect considerable discrepancy from each other, especially in colder conditions. Because parameterized $K$eq values are only dependent on ambient temperature, they continuously increase with time due to the decrease of temperature. In addition to discrepancy between different $K$eq parameterizations, $\varepsilon$ value varies dissimilarly with each $K$eq, ranging from 70% to 90%. All these results demonstrate that, in most cases, $K$eq values simply derived from existing database would fail to reproduce accurate relationship between $\mathrm{NO_3}$ and $\mathrm{N_2O_5}$.

To further elucidate the impact of $K$eq on deriving $\gamma(\mathrm{N_2O_5})$ via steady state approximation (hereafter defined as $\gamma_{\mathrm{ss}}(\mathrm{N_2O_5})$), Figure S6 shows the steady state fit based on all six database-derived $K$eq and in the same time periods as Figure S5 through Eq. (4) and Eq. (5) respectively (both of equations can derive a pair of $\gamma_{\mathrm{ss}}(\mathrm{N_2O_5})$ and $k\mathrm{NO_3}$). The $K$eq (corrected with $\varepsilon$) is calculated with $\mathrm{NO_3}$ and $\mathrm{N_2O_5}$ concentration simulated based on RACM2. Fit based on Eq. (4)

could lead to 11~46% underestimation of $\gamma_{ss}(N_2O_5)$, as indicated by varying slopes in Figure
S7(b)&(d), when using the database-derived $K$eq. Conversely, fit by Eq. (5) (shown in Figure
S7(a)&(c)) bias the result of $k$NO$_3$ served as the slopes without much influence on $\gamma_{ss}(N_2O_5)$
served as the intercept. Previous research ascribed inconsistent fit results between two
equations to measurements uncertainty (Brown et al., 2009;Brown et al., 2006). However, fit
with original $K$eq might be the primary reasons for such inconsistent results, and even deviates
the derived $\gamma_{ss}(N_2O_5)$ and $k$NO$_3$ from true values. Therefore, steady state fit based on Eq. (5)
might be the best choice for $\gamma(N_2O_5)$ derivation via steady state approximation. Similarly, Eq.
(4) is preferred to be applied when $k$NO$_3$ is the final objective.

## 3.2 Impacts of NO$_3$-N$_2$O$_5$ reactivity on steady state

In order to further explore the impacting factors on steady state fit method, $\gamma_{ss}(N_2O_5)$ results
are derived for each 2-hour time period of PKU2017 and TZ2018 dataset based on output from
steady state model. Since the pre-set $\gamma(N_2O_5)$ in this model is 0.02, the degree of deviation
from this value is supposed to reflect the accuracy of the fitted result.
It can be noticed from Eq. (5) that the variability of $k$NO$_3$ during the same time period
leads data points to scatter on lines with different slopes, which could bias the resulted $\gamma_{ss}(N_2O_5)$
from model pre-set value. As is shown in Figure 3, the absolute percentages of $\gamma_{ss}(N_2O_5)$
deviation grow dramatically with the increase of relative standard deviation of $k$NO$_3$ ($k$NO$_3$
RSD) in both of winter and summer data sets. The positive correlation even gives rise to
extreme deviation in summer data set with up to almost 10 times of model setting $\gamma(N_2O_5)$. In
fact, there remains accurate $\gamma_{ss}(N_2O_5)$ values derived in each range of $k$NO$_3$ RSD, indicating a
not strictly positive correlation between $\gamma_{ss}(N_2O_5)$ deviation and $k$NO$_3$ RSD. It implies that
large variation of $k$NO$_3$ only enhance the possibilities of inaccurate results from steady state
fit rather than hinder the $\gamma_{ss}(N_2O_5)$ quantification all the time.

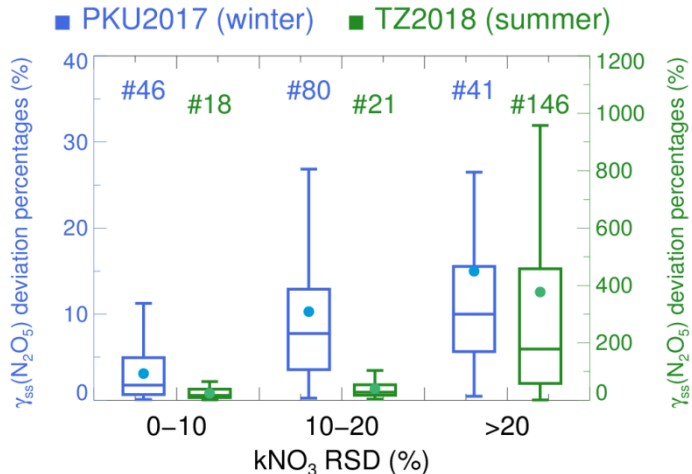


**Figure 3.** Relationship between $\gamma(N_2O_5)$ derivation through steady state approximation and $k$NO$_3$ relative standard deviation (RSD) in box whisker plot. The blue and green color represent dataset from PKU2017 and TZ2018 respectively, binned according to $k$NO$_3$ RSD. The dots are the mean deviation of $\gamma_{ss}(N_2O_5)$. The number above the box whisker represents the valid data points in each bin.

Besides the large variation of $k$NO$_3$ in short time period, the absolute level of $k$NO$_3$ and $k$N$_2$O$_5$ could influence the possibilities of inaccurate $\gamma_{ss}(N_2O_5)$ from different aspects. Although the enhancement of $k$NO$_3$ and $k$N$_2$O$_5$ boost the approach to steady state (Text. S5 and Figure S8), higher levels of $k$NO$_3$ amplify the bias of $\gamma_{ss}(N_2O_5)$, contrary to $k$N$_2$O$_5$, with the same relative variation of $k$NO$_3$ (Text. S6 and Figure S10). It indicates that the region with plural emissions (e.g. strong biogenic or vehicular emission) might not be suited for steady state fit due to the high $k$NO$_3$. Therefore, a trade-off between the variation of $k$NO$_3$ and the high level of $k$NO$_3$ (fast approach to steady state) should be made when derive $\gamma_{ss}(N_2O_5)$.

## 3.3 Implication for accurate steady state analysis of NO$_3$-N$_2$O$_5$

While a few studies have examined the validity of steady state under certain conditions via numerical modeling when interpreted the ambient data (Brown et al., 2009;Brown et al., 2003), a clear range well suited to steady state analysis of NO$_3$-N$_2$O$_5$, taking both $K$eq and validity of steady state into consideration, has not been determined to date.

Here almost 20000 simulations are displayed in the parallel plot of Figure 4, where each line connects 5 constraint parameters to the calculated steady state time and $\varepsilon$ (the correction factor for $K$eq parameterization to match the exact ratio of [N$_2$O$_5$]/([NO$_2$]$\times$[NO$_3$]), detailed in Eq.6). The gray traces represent the simulations could not match steady state within 600 s and were defined as less valid cases here. By this definition, we intend to indicate that it is also viable to apply steady state approximation on air mass, which requires more than 600 s to match steady state, whereas the uncertainty caused therefrom could increase to some extent. The pink and blue traces together represent the simulations could match valid steady state within 600 s without consideration of $K$eq deviation (in other word the value of $\varepsilon$). Furthermore, the criterion to apply steady state approximation appropriately we defined is that approach to steady state within 600 s and the $\varepsilon$ larger than 0.9, which are indicated as pink traces. While the level of $T$, NO$_2$ and O$_3$ have minor effect on the approach to steady state, simultaneous low $k$N$_2$O$_5$ (indicated as low Sa in the plot) and $k$NO$_3$ prevent the NO$_3$-N$_2$O$_5$ system from developing steady state. For example, when $k$NO$_3$ is lower than 0.01 s$^{-1}$, the air mass will be valid only if Sa increases to at least 3000 $\mu m^2\,cm^{-3}$ with $\gamma(N_2O_5)$ of 0.02. It implies that clean air mass is not suited for steady state in any cases, whereas high aerosol condition provides more possibilities to approach steady state even with low $k$NO$_3$. However, in order

to interpreting $NO_3$-$N_2O_5$ chemistry with accurate $K$eq coefficient, the ε larger than 0.9 is
additionally taken into consideration, which excludes 50% of valid steady state cases mainly
with high aerosol and lower than 10°C. These cases could bias $[N_2O_5]/([NO_2]\times[NO_3])$ from
original $K$eq (also indicated in Figure 2), leading to inaccurate results of calculation based on
$K$eq.

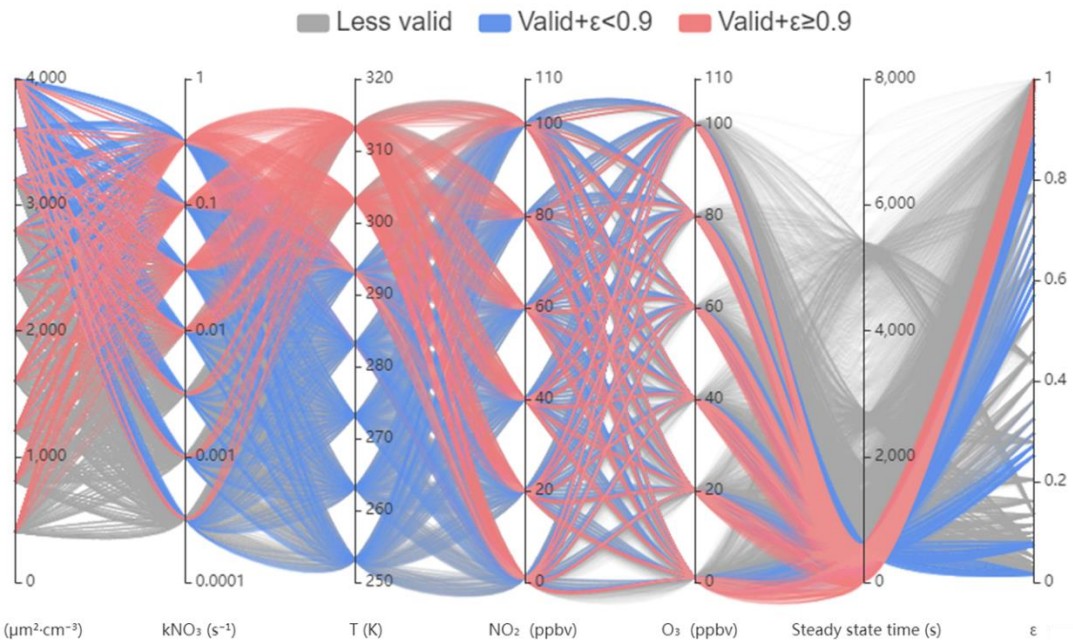


**Figure 4.** Numerical simulations for determining conditions available for steady state approximation

method in a parallel axis plot. Each line simply represents a simulation associated with different parameters
in different vertical axes. The first five axes from the left represent initial variables used for constraining
the simulations respectively. The last two axes represent the time required for achieving steady state and the
ε value calculated from the simulated results. The gray lines show cases approaching steady state longer
than 600 s (less valid). The blue lines show cases approaching steady state cases within 600 s while with ε
less than 0.9, which is also inappropriate for steady state analysis. The pink lines show cases approaching
steady state cases within 600 s with ε higher than 0.9, which is suited for steady state analysis.

## 4 Conclusions

In this study, we found that the parameterized $K$eq coefficient deviates much from the ratio of
$[N_2O_5]/([NO_2]\times[NO_3])$ in some cases where steady state is valid. The indicator of the deviation,
ε, is relatively sensitive to $N_2O_5$ reactivity and ambient temperature. It implies that conditions
suited for steady state analysis should be determined according to not only the validity of
steady state but also $K$eq especially under high aerosol conditions, like some regions in India,
China, Europe and the US (Baasandorj et al., 2017;Cesari et al., 2018;Huang et al.,
2014;Mogno et al., 2021;Petit et al., 2017;Wang et al., 2017b). Considering that high level of
$k\text{NO}_3$ might amplify the bias of $\gamma_{ss}(\text{N}_2\text{O}_5)$ yield from steady state fit and appears to be
accompanied with fast variations, air mass of $k\text{NO}_3$ less than 0.01 s$^{-1}$ with high aerosol and $T$
higher than 10°C is therefore the best suited for steady state analysis of $\text{NO}_3$-$\text{N}_2\text{O}_5$ chemistry,
which indicates that this method would be more applicable in polluted regions with high
aerosol loading during summertime. If the restriction of $\varepsilon$ is relaxed to 30%, some of winter
conditions will also be applicable. Our results provide an insight to improve the accuracy of
steady state approximation method and find suited areas to interpret nighttime chemistry.
Further improvement of in-situ $\text{NO}_3$-$\text{N}_2\text{O}_5$ budgets quantification might relies on the direct
measurements via flow tube system or machine learning prediction based on ancillary
parameters.

**Supporting Information:** The Supporting Information is available on line.

**Code/Data availability.** The datasets used in this study are available from the corresponding
author upon request (wanghch27@mail.sysu.edu.cn; k.lu@pku.edu.cn).

**Author contributions.** K.D.L. and H.C.W. designed the study. X.R.C and H.C.W. analyzed
the data and wrote the paper with input from K.D.L.

**Competing interests**. The authors declare that they have no conflicts of interest.

**Acknowledgments**. This project is supported by the National Natural Science Foundation of
China (21976006, 42175111); the Beijing Municipal Natural Science Foundation for
Distinguished Young Scholars (JQ19031); National State Environmental Protection Key
Laboratory of Formation and Prevention of Urban Air Pollution Complex (CX2020080578);
the special fund of the State Key Joint Laboratory of Environment Simulation and Pollution
Control (21K02ESPCP); the National Research Program for Key Issue in Air Pollution
Control (DQGG0103-01, 2019YFC0214800). Thanks for the data contributed by field
campaign team.

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
