# Peer review of "Interpretation of NO3-N2O5 observation via steady state in high aerosol air"

_Atmospheric Chemistry and Physics, 2021_

## Author Comment (AC1)

**Response to Editors and Reviewers**

We appreciate the reviewers for their careful reading and their constructive comments on our manuscript. As detailed below, the reviewer's comments are shown as *italicized font*, our response to the comments are normal font. New or modified text is in blue.

All of the line numbers refer to Manuscript ID: acp-2021-886.

Reviewer #1

*This work assesses the neglected uncertainty in steady state approximation for interpreting NO3 and N2O5 based on in-situ observation data and explores the key influencing factors for the accuracies of equilibrium coefficient and heterogeneous uptake coefficient. The results highlight the large impacts of aerosol loading, NO3 reactivity, and ambient temperature on the chemical reaction coefficients and provide a good solution for performing accurate steady state approximation in particular in high aerosol loading conditions. The manuscript is generally well written, with innovative methods, deep mechanism investigation, full discussion, and fluent language. It can be considered to accept after addressing the following minor comments.*

1. *Line 27-28, can the "concentration ranges appropriate" be "appropriate concentration ranges"?*

The statement is modified as suggested.

2. *Line 130, removal the "0" before "1, 2, 3, 4" in the labels of the horizontal axis.*

   The figure is modified as suggested.

3. *Line 222, specify the "plural emissions", e.g., strong biogenic or vehicle emissions.*

   Thank you for the suggestion. We modify the description as follows.

   "It indicates that the region with plural emissions (e.g. strong biogenic or vehicular emission) might not be suited for steady state fit due to the high $k\mathrm{NO_3}$."

4. *Line 231, suggest pointing out the meaning of the ε, e.g., the correction factor for [N2O5]/ ([NO2]×[NO3]), when it appears for the first time in a new section.*

   Thank you for the suggestion. We add the meaning of this parameter here for better clarification as follows.

   "Almost 20000 simulations are displayed in the parallel plot of Figure 4, where each line connects 5 constraint parameters to the calculated steady state time and ε (the correction factor for Keq parameterization to match the exact ratio of $[\mathrm{N_2O_5}]/([\mathrm{NO_2}]\times[\mathrm{NO_3}])$, detailed in Eq.6)."

---

## Author Comment (AC2)

Reviewer #2

**General comments:**

*Chen et al. analyze the steady state approximation in the interpretation of observations of NO₃ and N₂O₅ mixing ratios in ambient air with a focus on the potential error arising from the deviations in the equilibrium constant, and the resulting potential sensitivity of N₂O₅ uptake coefficients derived from this method. They use data from two recent ground-based field campaigns in and near Beijing. Of particular value is the assessment of the equilibrium constant to the actual reaction quotient (i.e., Keq compared to [N₂O₅]/([NO₂][NO₃])). This effect results from the competition between reactive uptake for N₂O₅ and thermal dissociation of N₂O₅ in the equilibrium, and should be described as such (see comments below). The variation in equilibrium constants from different databases or reaction mechanisms is also particularly valuable, and does not appear elsewhere in the literature to my knowledge.*

*The analysis will be of interest to the atmospheric chemistry community generally and those who study nighttime chemistry specifically. I recommend publication after attention to the following comments.*

*The most general overall comment for this manuscript in clarity. This is true in a number of places. For example, equations (1) and (2) define the equations used in the analysis, but the equations themselves are not derived, their assumptions not justified, and the chemistry that underlies them not defined. The introduction should presume the reader do not have an intimate familiarity with nighttime chemistry and should take the time to define assumptions and chemistry. For example, what exactly is kNO3, what controls it and what values should it have? The oxidation chemistry that leads to this loss rate constant should be described, at least briefly.*

Thank you for your suggestion. We add the statement to clarify how to derive the steady state equation, formerly shown as Eq (1)&(2), in the introduction and to justify the assumptions and chemistry underlying them in the method section. The definition of $k$NO₃, the factors impacting this parameter, the typical values in real atmosphere and the oxidation chemistry lead to this loss rate are also briefly described at the first time it appears. The modifications are as follows.

"Under the steady state condition, the lifetime of NO₃ (denoted as $\tau_{ss}(\mathrm{NO_3})$) can be calculated as the ratio of NO₃ concentration over the production rate

($k_{NO_2+O_3}[\mathrm{NO_2}][\mathrm{O_3}]$) or over the removal rate of both NO₃ and N₂O₅, as indicated in

Eq. (1). A similar representation of N₂O₅ steady state lifetime is also shown in Eq. (2). The loss frequencies of various sink pathways of NO₃ and N₂O₅ are integrated as total first-order in the following equations, represented by $k$NO₃ and $k$N₂O₅ term. Briefly, the kNO₃ is contributed by the reaction of NO₃ radical with NO and hydrocarbons and uptake on particles at night, ranging from hundredths of s⁻¹ to several s⁻¹ depending on the air mass. Due to its large rate constant with NO, the concentration usually

dominates the lifetime of $NO_3$ radical in urban areas with fresh NO emission. Otherwise, the reactions with hydrocarbons, especially unsaturated hydrocarbons, is preferential for $NO_3$ in rural areas. The $K$eq denotes the equilibrium coefficient for reactions R1a and R1b, used to be derived by Eq. (3).

$$\tau_{ss}(NO_3) \equiv \frac{[NO_3]}{k_{R1}[NO_2][O_3]} \approx (k_{NO_3} + K_{eq}[NO_2]k_{N_2O_5})^{-1}, \tag{1}$$

$$\tau_{ss}(N_2O_5) \equiv \frac{[N_2O_5]}{k_{R1}[NO_2][O_3]} \approx (k_{N_2O_5} + \frac{k_{NO_3}}{K_{eq}[NO_2]})^{-1}, \tag{2}$$

$$K_{eq} = \frac{k_{R1a}}{k_{R1b}} = \frac{[N_2O_5]}{[NO_2][NO_3]}, \tag{3}"$$

"The framework of steady state approximation for $NO_3$-$N_2O_5$ system is basically built on its chemical production and removal pathways, in case of extremely weak physical processes relative to its chemical processes. With simultaneous measurements of $NO_3$, $N_2O_5$ and relevant precursor concentrations, the steady state lifetime $\tau_{ss}(NO_3)$ and $\tau_{ss}(N_2O_5)$ can be quantified for a targeted period as shown in Eq. (1) and Eq. (2). By substituting the $kN_2O_5$ with $0.25 \times c \times S_a \times \gamma(N_2O_5)$, the $\gamma(N_2O_5)$ and the reactivity of $NO_3$ ($kNO_3$, including the reactions of $NO_3$ with NO and hydrocarbons) can therefore be determined by Eq. (4) and Eq. (5).

$$\tau_{ss}^{-1}(NO_3) \approx k_{NO_3} + 0.25cS_aK_{eq}[NO_2]\gamma(N_2O_5), \tag{4}$$

$$(0.25cS_a\tau_{ss}(N_2O_5))^{-1} \approx \gamma(N_2O_5) + k_{NO_3}(0.25cS_aK_{eq}[NO_2])^{-1}, \tag{5}$$

Here $c$ represents the mean molecular velocity of $N_2O_5$, Sa represents the aerosol surface area and the $K$eq is calculated from the rate constant of reversible reactions R1a ($k_{R1a}$) and R1b ($k_{R1b}$), which is a temperature-dependent parameter. It should be noted that the photolysis of $NO_3$ is not considered in the $kNO_3$ due to weak radiation at night and the homogeneous hydrolysis was also ignored due to its small contribution in comparison to heterogeneous pathway, similar presumption was also implemented in previous studies (Brown et al., 2009;Mentel et al., 1996;Wahner et al., 1998). In the form of these two equations, the potential covariance between Sa and $NO_2$ concentration can be avoided to decrease the uncertainty (Brown et al., 2009). By fit to these two equations, $\gamma(N_2O_5)$ can be directly derived from slope of the plot of $\tau_{ss}^{-1}(NO_3)$ against $0.25cS_aK_{eq}[NO_2]$ or from intercept of the plot of $(0.25cS_a\tau_{ss}(N_2O_5))^{-1}$ against $(0.25cS_aK_{eq}[NO_2])^{-1}$ respectively."

*Other examples are given in the specific comments below.*

**Specific comments:**

*1.   Line 39: The steady state approximation dates well before Brown 2003, to the early work of Platt and coworkers, e.g.   Platt, U., D. Perner, J. Schräder, C. Kessler, and A. Toennissen, The Diurnal Variation of NO₃. J. Geophys. Res., 1981. **86**(C12): p. 11965-11970.*

We add this reference into the text here.

*2.   Line 45: Text S1 and Figure S1 are referenced at this point, although the methods used in here have not yet been introduced.   For example, the scatter plots at the bottom of Figure S1 have rather complicated axes, and the slopes are stated as being a measure of the N₂O₅ uptake coefficient.   Equations describing these relationships should proceed their presentation in a figure.*

We add brief descriptions of the purpose of linear fit in the caption of Figure S1 and in text S1 where Figure S1 is referenced. The modification is as follows.

"**Figure S1.** Exemplary steady state fit and the variations of relevant parameters in ambient conditions of (a)&(b)&(c)&(d) PKU site and (e)&(f)&(g)&(h) TZ site. The red dots in (d)&(h) represent the correlation plot between $\left(0.25cS_a\tau_{ss}(N_2O_5)\right)^{-1}$ and $\left(0.25cS_aK_{eq}[NO_2]\right)^{-1}$ used for deriving $\gamma(N_2O_5)$ and $kNO_3$ as illustrated in the method. The text on the plot gives the best fit results of $\gamma(N_2O_5)$ and correlation coefficient."

"Over the period of wintertime case shown in Figure S1, the NOx and Sa concentration were low, indicating a clean episode. The $\gamma(N_2O_5)$ and $kNO_3$ can be determined from the intercept and slope respectively by linear fit based on steady state equation Eq. (5). The details and derivation of this approach are provided in introduction and method section."

*3.   Line 57: Reference to iterative box model is not defined.   Those who have worked with such a model will understand this statement, but it will not be clear to most readers.*

We provided brief statements here to explain the iterative box model as reviewer suggested. The modification is as follows.

"In addition, the mathematical conversion between $NO_3$ and $N_2O_5$ concentration via Keq coefficient can simplified the calculation in the iterative box model, which derives $\gamma(N_2O_5)$ by iterating its value in the model until the predicted $N_2O_5$ concentration matches the observation (Wagner et al., 2013;Wang et al., 2020b)."

*4.   Line 62: The phrase "half artificial" is not defined.   A reference to the following sections (e.g., "see below") is needed at a minimum.*

Thank you for the suggestion and the modified statement is as follows.

"In this study, we formulate a half artificial dataset with expected properties based on field campaigns. Specifically, most of species contained in the dataset are observed

values while only $NO_3$ and $N_2O_5$ were calculated by the steady state model (illustrated in the section 2.2)"

5.   *Line 64: Linear fit between which variables?   Does this refer to equations 1 and 2 that follow?   Specify if this is the case.*

The linear fit is conducted according to Eq. (4) and Eq. (5) after revised and the variables among them are denoted in section 2.1 as follows.

"By fit to these two equations, $\gamma(N_2O_5)$ can be directly derived from slope of the plot of $\tau_{ss}^{-1}(NO_3)$ against $0.25cS_aK_{eq}[NO_2]$ or from intercept of the plot of

$(0.25cS_a\tau_{ss}(N_2O_5))^{-1}$ against $(0.25cS_aK_{eq}[NO_2])^{-1}$ respectively. In the following analysis, the linear fit based on Eq. (5) is preferred in steady state approximation."

We then modify the statement here to specify the meaning of linear fit as follows.

"With the dataset, we illustrate the reasons for deviation of parameterized $K$eq from $[N_2O_5]/([NO_2]\times[NO_3])$ in ambient conditions, the possible uncertainties of linear fit based on steady state equations Eq. (4) and Eq. (5) (the related variables are explained in section 2.1) resulted from different $K$eq"

6.   *Line 72: "steady" rather than "stead".   This error appears in multiple places.   Authors should search on "stead" and replace with "steady".*

Thanks for your help on correction. We check the similar errors and modify them throughout the manuscript.

7.   *Line 73: Does weak physical processes mean transport and boundary layer dynamics or something else?   Specify.*

The weak physical processes mean weak transport under low wind speed, weak dilution from boundary layer dynamics at night and slow deposition in case of no rain. We specified the physical processes as follows.

"The framework of steady state approximation for $NO_3$-$N_2O_5$ system is basically built on its chemical production and removal pathways, in case of extremely weak physical processes (e.g. transport, dilution and deposition) relative to its chemical processes."

8.   *Line 78-79, equations 1 and 2: These equations are complicated and lack a statement of assumptions or a derivation.   The text should provide these.*

Thank you for pointing out this deficiency. As per our reply to general comments above, we add the equation derivation in the introduction as indicated by Eq. (1)~(3) in a revised version and provided the statement of assumption in section 2.1 to support the linear fit by Eq (4) & (5). The modifications are as follows.

"Under the steady state condition, the lifetime of $NO_3$ (denoted as $\tau_{ss}(NO_3)$) can be calculated as the ratio of $NO_3$ concentration over the production rate

$(k_{NO_2+O_3}[NO_2][O_3])$ or over the removal rate of both $NO_3$ and $N_2O_5$, as indicated in

Eq. (1). A similar representation of $N_2O_5$ steady state lifetime is also shown in Eq. (2). The loss frequencies of various sink pathways of $NO_3$ and $N_2O_5$ are integrated as total first-order in the following equations, represented by $kNO_3$ and $kN_2O_5$ term. Briefly, the $kNO_3$ is contributed by the reaction of $NO_3$ radical with NO and hydrocarbons and uptake on particles at night, ranging from hundredths of $s^{-1}$ to several $s^{-1}$ depending on the air mass. Due to its large rate constant with NO, the concentration usually dominates the lifetime of $NO_3$ radical in urban areas with fresh NO emission. Otherwise, the reactions with hydrocarbons, especially unsaturated hydrocarbons, is preferential for $NO_3$ in rural areas. The $K$eq denotes the equilibrium coefficient for reactions R1a and R1b, used to be derived by Eq. (3).

$$\tau_{ss}(NO_3) \equiv \frac{[NO_3]}{k_{R1}[NO_2][O_3]} \approx (k_{NO_3} + K_{eq}[NO_2]k_{N_2O_5})^{-1}, \qquad (1)$$

$$\tau_{ss}(N_2O_5) \equiv \frac{[N_2O_5]}{k_{R1}[NO_2][O_3]} \approx (k_{N_2O_5} + \frac{k_{NO_3}}{K_{eq}[NO_2]})^{-1}, \qquad (2)$$

$$K_{eq} = \frac{k_{R1a}}{k_{R1b}} = \frac{[N_2O_5]}{[NO_2][NO_3]}, \qquad (3)"$$

"With simultaneous measurements of $NO_3$, $N_2O_5$ and relevant precursor concentrations, the steady state lifetime $\tau_{ss}(NO_3)$ and $\tau_{ss}(N_2O_5)$ can be quantified for a targeted period as shown in Eq. (1) and Eq. (2). By substituting the $kN_2O_5$ with $0.25 \times c \times S_a \times \gamma(N_2O_5)$, the $\gamma(N_2O_5)$ and the reactivity of $NO_3$ ($kNO_3$, including the reactions of $NO_3$ with NO and hydrocarbons) can therefore be determined by Eq. (4) and Eq. (5).

$$\tau_{ss}^{-1}(NO_3) \approx k_{NO_3} + 0.25cS_aK_{eq}[NO_2]\gamma(N_2O_5), \qquad (4)$$

$$\left(0.25cS_a\tau_{ss}(N_2O_5)\right)^{-1} \approx \gamma(N_2O_5) + k_{NO_3}\left(0.25cS_aK_{eq}[NO_2]\right)^{-1}, \qquad (5)$$

Here $c$ represents the mean molecular velocity of $N_2O_5$, $S_a$ represents the aerosol surface area and the $K$eq is calculated from the rate constant of reversible reactions R1a ($k_{R1a}$) and R1b ($k_{R1b}$), which is a temperature-dependent parameter. It should be noted that the photolysis of $NO_3$ is not considered in the $kNO_3$ due to weak radiation at night and the homogeneous hydrolysis was also ignored due to its small contribution in comparison to heterogeneous pathway, similar presumption was also implemented in previous studies (Brown et al., 2009;Mentel et al., 1996;Wahner et al., 1998)."

*9. Line 100-107: The description of the artificial data set is not clear. First, the justification for the term "half artificial" is not clearly defined – i.e, what about this is only half artificial (or half real) rather than completely artificial or real? Second, it seems that the authors are applying a test for the validity of steady state and then*

*excluding data points that do not meet a steady state criterion (itself not defined … is this agreement of model with observed NO₃ or N₂O₅ to within the stated limits, or is this just model to model?). The text reads, however, as though large amounts of data are simply being excluded from the analysis or replaced with artificial data (as stated further down in the paragraph) if they do not fit the model, which would be neither scientifically rigorous nor consistent with what is shown in the figures. Possibly the issue here is with the text-based description. The authors should read this carefully to be sure that it means what they intend.*

Thank you for the comment. For the first question, the reason why we defined this dataset as half artificial is that most of species contained in it are observed values while the $NO_3$ and $N_2O_5$ were predicted by the steady state model (illustrated in the section 2.2), from which each data point of $NO_3$ and $N_2O_5$ concentration were outputted in steady state as far as possible after filtering. We then combine the simulated values ($NO_3$ and $N_2O_5$) and the observed values to generate this half artificial dataset for further analysis. More clarification in the main text are added as suggested. For the second question, whether each data point satisfied steady state is validated according to the steady state criterion, defined by the deviation between steady state lifetime of $N_2O_5$ ($\tau_{ss}(N_2O_5) = \frac{[N_2O_5]}{k_{R1}[NO_2][O_3]}$) and calculated lifetime of

$N_2O_5$ ($\tau_{calc}(N_2O_5) = (k_{N_2O_5} + \frac{k_{NO_3}}{K_{eq}[NO_2]})^{-1}$). If the deviation exceeds 10% for a data point, it will be excluded from the following analysis. In addition, the data higher than 5 ppbv NO is also filtered out in the following calculation, as it causes large variability of $kNO_3$ to bias the linear fit though the $NO_3$ and $N_2O_5$ approach the steady state rapidly under high NO. The fraction of excluded data is less than 8%, which are expected to have little influence on our results.

We carefully revised the text describing this half artificial dataset to make it clear to readers. The text after modification is as follows.

"The steady state model is reformed from 0-dimension box model to produce $NO_3$ and $N_2O_5$ which are in steady state as far as possible. It is constrained by measurements of NO, $NO_2$, $O_3$, CO, $CH_4$, VOCs, HCHO, Sa, relative humidity (RH), temperature (T), pressure, coupled with Regional Atmospheric Chemistry Mechanism, version 2 (RACM2). Each data point is treated as an independent air mass, aging 10 hours and keeping input constraint unchanged. As $NO_3$-$N_2O_5$ chemistry, the interest of this work, usually shows marked impacts during the night, only the time periods with negligible photolysis frequency are under consideration. In the standard simulation (herein referred as Mod0), the uptake coefficient of $N_2O_5$ is set to 0.02, as a reasonable value of literatures (Brown et al., 2006;Chen et al., 2020;McDuffie et al., 2018;Morgan et al., 2015;Phillips et al., 2016;Wagner et al., 2013;Wang et al., 2017c;Yu et al., 2020).

Two half-artificial datasets are derived from PKU2017 and TZ2018 field campaigns (see Text S2) based on steady state model for analysis in the following sections. The

simulated $NO_3$ and $N_2O_5$ and other observed values used for the constraints of steady state model jointly formulate these half-artificial datasets. Specifically, the $NO_3$ and $N_2O_5$ concentration in this dataset are the output of the steady state model simulation, and guaranteed to be in steady state with respect to other observed precursors. To verify the steady state of $NO_3$ and $N_2O_5$ for each data point, we filtered the data set according to deviation between steady state lifetime of $N_2O_5$ ($\tau_{ss}(N_2O_5) = \frac{[N_2O_5]}{k_{R1}[NO_2][O_3]}$) and calculated lifetime of $N_2O_5$ ($\tau_{calc}(N_2O_5) = (k_{N_2O_5} + \frac{k_{NO_3}}{K_{eq}[NO_2]})^{-1}$).

If the deviation exceeds 10% for a data point, it will be excluded from the following analysis. We presume that if any data point outputted from the model is still out of steady state in terms of $NO_3$ and $N_2O_5$, the sink rate constant of air mass represented by this data point should be too weak for steady state analysis within a reasonable timescale. In addition, the data higher than 5 ppbv NO is filtered out in the following calculation, since the resulting large variation of $kNO_3$ can bias the linear fit even though the $NO_3$ and $N_2O_5$ approach the steady state rapidly under high NO (discussed in 3.2). The fraction of excluded data is less than 8%, which are expected to have little influence on our results. The calculated nighttime loss fraction accounted by $NO_3$ and $N_2O_5$ show large discrepancy (see Text. S3 and Figure S2) between these two half-artificial datasets, which provide us a good opportunity to investigate the impacting factors on steady state approximation across different conditions.

Rather than using observation data directly, a half-artificial dataset can provide larger amount of valid data for steady state analysis with known $\gamma(N_2O_5)$ value. Besides, this method avoids the impacts from steady state deviation, which helps to analyze the factors influencing $\gamma(N_2O_5)$ quantification via steady state approximation backwards from a known steady state condition."

*10. Line 108: 5 ppbv NO is very large indeed, and is equivalent to an $NO_3$ loss rate coefficient of approximately 3 s$^{-1}$, likely much larger than rate coefficients associated with $N_2O_5$ uptake or $NO_3$-VOC reactions. Is this a sufficient filter for assessing $NO_3$ chemistry that is not attributable to NO reaction?*

Thank you for the comment. The reason why we excluded data with higher than 5 ppbv NO is that this part of data has extremely large $NO_3$ loss rate coefficient just as you mentioned and could considerably influence the linear fit based on steady state approximation. In fact, we did not intend to eliminate the influence resulted from NO entirely. Instead, small variation of NO would allow for assessments on the accuracy of steady state approximation over a range of $kNO_3$ in this work, although it will hinder the analysis of $NO_3$ chemistry in real atmosphere. Furthermore, most of data would not be excluded under this limit. Therefore, the limit of 5 ppbv NO made a balance between excluding influence from large $kNO_3$ variance and retaining most of data for analysis.

*11. Figure 1: What starting mixing ratios of $NO_2$ and $O_3$ are used in this simulation? Are these held constant? Are the results different if these are initialized and then allowed to vary? The $N_2O_5$ loss rate constant is given in the figure, but*

*should be compared explicitly to the thermal lifetime (thermal dissociation rate constant) of N$_2$O$_5$ in the equilibrium. Does this ratio match that of the incomplete equilibrium shown in panel D?*

The starting mixing ratios of NO$_2$ and O$_3$ are 10 and 23 ppbv respectively in this simulation, which is the average level for the nighttime conditions in PKU2017. They are held constant in the simulation. This result will stay almost the same if these starting values are initialized and allowed to vary (show in the figure below). We further add some clarification on this simulation in the main text as follows.

"The NO$_3$-N$_2$O$_5$ achieves steady state after 1.5-hours evolution, when concentration and rates remain constant (Figure 1). In this simulation, the starting mixing ratios of NO$_2$ and O$_3$ are 10 and 23 ppbv respectively, which is the average level for the nighttime conditions in PKU2017. The concentration of these two precursors are held constant in the simulation to better illustrate the influence of removal rates. This result will stay almost the same if these starting values are initialized and allowed to vary."

[Figure]

As suggested by the reviewer, we compare the loss rate constant N$_2$O$_5$, indicated in panel (a), to its thermal dissociation rate constant. The exact velocity of N$_2$O$_5$ thermal dissociation is also shown as the red dash line in panel (d). We found that this ratio is different from the factor $\varepsilon$ in panel (d) but is slightly higher than 1-$\varepsilon$, which we presume might represent the ratio of incomplete equilibrium mentioned by the reviewer.

*12. Line 146: Losses for NO$_3$ and N$_2$O$_5$ are generally not equal in creating a deviation from equilibrium. Large NO$_3$ loss rate constants normally do not lead to a deviation from equilibrium, whereas even relatively modest N$_2$O$_5$ rate constants can*

*easily lead to large deviations from equilibrium. This is the result of competition between N₂O₅ loss through aerosol uptake and thermal dissociation, where the thermal dissociation rate constant is slow at colder temperatures and therefore is subject to competition reactions. Rapid reaction of NO₃, by contrast, generally does not compete with the rapid forward reaction. Suggest removing the reference to NO₃ loss here.*

Thank you for your suggestion. We therefore remove the reference to NO₃ loss in the text.

*13. Figure 2: The relationship of the top and bottom axes are not clear. One would expect that the change in kNO3 or kN2O5 would correspond directly to the change in temperature on the top axis, but this appears not to be the case. Therefore, it is not clear what the lines on the plot are showing – i.e., the influence of temperature or the rate constants. Possibly different traces are plotted against different axes, but this is not specified anywhere in the text. Clarify the description or separate these plots into two different sets. Also, as per the previous comment, any dependence of the deviation from equilibrium on either temperature or N₂O₅ loss rate constant would co-vary, so the condition chosen for either T or kN2O5 in the complementary plot needs to be specified. This is not specified in the text or the figure caption.*

Thank you for pointing out the confusing traces and axes displayed here. Here we plotted the trace of temperature on the upper horizontal axis while the other two on the lower axis. After modification, we clarify the description of this configuration in the figure caption. For the second question, the condition chosen for the dependency plots are listed in table S2 and we further specify it in the figure caption after modification.

The figure2 caption is modified as follows.

"**Figure 2.** Sensitivity plot of $k\text{NO}_3$, $k\text{N}_2\text{O}_5$ and Temperature (T) against coefficient ε. The trace of T is plotted against the upper horizontal axis and the traces of the other two parameters are plotted against the lower horizontal axis. (a) Basic model condition is according to typical winter condition of PKU2017; (b) Basic model condition is according to typical summer condition of TZ2018. Basic model conditions including $k\text{NO}_3$, $k\text{N}_2\text{O}_5$ and Temperature (T) are shown in Table S2. It should be noted that the provided ranges of each factor do not exactly equal to but encompass the ambient conditions encountered during the two campaigns."

*14. Line 183, Figure S6: Give the values of the derived gamma_N2O5 – one can see the differences in the slopes of the lines in the figures, but the quantitative values should also be given on the figures or in a separate table.*

We add the values of derived $\gamma(\text{N}_2\text{O}_5)$ in the figure S6 (attached below).

[Figure]

*15. Line 230-244 and Figure 4: The parallel axis plot in Figure 4 is visually appealing, but difficult to interpret. What is plotted along the bottom of the figure, and why do lines connecting each of the axes have curvature along the bottom axis? Is each line simply a collection of simulations associated with a given parameter in each axis but different parameters in the other axes? It is not clear from the text exactly how this plot has been constructed.*

Thank you for pointing out the confusion. For the construction of the figure, no horizontal axis these traces are plotted along and each line simply represent a simulation associated with different parameters in different vertical axes. For the curvature along the bottom, we intended to avoid the overlap of traces in different colors by connecting each parameter in curve lines. Therefore, there is no exact meaning of these curvature along the bottom, since they are resulted from the way we plotted the traces. We further specify it in the figure caption.

*16. What is clear is that the grey lines indicate invalid steady state, defined as an approach time greater than 600 s. This appears arbitrary, as observed NO₃ and N₂O₅ are often much more aged than 10 minutes from the point of emission of NOₓ or the time since sunset. The authors may want to consider a different definition with a longer time horizon, or if not, the statement about 600 s should not be simply "invalid" but rather described as an arbitrary threshold.*

Thank you for the suggestion. We therefore changed the statement "invalid" to "less valid" in Figure 4 and the corresponding text to explain this definition.

By this modification, we intend to indicate that it is also viable to apply steady state approximation on air mass, which requires more than 600 s to match steady state. However, there might exist larger uncertainties by doing this than applying steady state approximation on air mass approaching steady state faster than 600 s. The modified text and figure are as follows.

"Here almost 20000 simulations are displayed in the parallel plot of Figure 4, where each line connects 5 constraint parameters to the calculated steady state time and ε (the correction factor for Keq parameterization to match the exact ratio of $[N_2O_5]/([NO_2] \times [NO_3])$, detailed in Eq.3). The gray traces represent the simulations could not match steady state within 600 s and were defined as less valid cases here. By this definition, we intend to indicate that it is also viable to apply steady state approximation on air mass, which requires more than 600 s to match steady state (less valid), whereas the uncertainty caused therefrom could increase to some extent. The pink and blue traces together represent the simulations could match valid steady state within 600 s without consideration of Keq deviation (in other word the value of ε). Furthermore, the criterion to apply steady state approximation appropriately we defined is that approach to steady state within 600 s and the ε larger than 0.9, which are indicated as pink traces."

[Figure]

**Supplement**

*17. Lines 62-65: NO values near the detection limit are corrected. What is this correction? Does this mean the data were arbitrarily adjusted to zero during periods when it is anticipated that they would be zero?*

We adjusted the NO value to zero when the $O_3$ level was higher than 25 ppbv, as the lifetime of NO would be extremely short at night under this condition and the NO measurement at the low level usually has large uncertainty. This adjustment will not change our results. A clarification was added here after revision as follows.

"We adjusted the nighttime NO concentration near the detection limit to zero during the periods with $O_3$ concentration higher than 25 ppbv, as the lifetime of NO would be

extremely short at night under this condition and the NO measurement at the low level usually has large uncertainty."

*18.  Line 88-90 and Figure S3: What is Tau(N₂O₅)SS and Tau(N₂O₅)Calc?  These quantities are not defined in the figure caption, the supplement text or the main text.  It is difficult to understand what is being compared here, especially given the remarkable agreement between what appears to be an observational and model quantity.*

Thank you for the comment. To verify the steady state of $NO_3$ and $N_2O_5$ for each data point, we made comparisons between steady state lifetime of $N_2O_5$ ($\tau_{ss}(N_2O_5) = \frac{[N_2O_5]}{k_{R1}[NO_2][O_3]}$) and calculated lifetime of $N_2O_5$ ($\tau_{calc}(N_2O_5) = (k_{N_2O_5} + \frac{k_{NO_3}}{K_{eq}[NO_2]})^{-1}$).

If the deviation between them exceeds 10% for a data point, it will be assumed as out of steady state. After the modification, we added the clarification of these parameters in the method (see our reply to comment 9) and supplement here as follows.

"Taking two typical cases from these two datasets for example, the $N_2O_5$ lifetime was about 20 minutes in winter (Figure S3(a)), while was largely reduced to 100 seconds for summertime case (Figure S3(c)). The steady state lifetime of $N_2O_5$ ($\tau_{ss}(N_2O_5) = \frac{[N_2O_5]}{k_{R1}[NO_2][O_3]}$) and calculated lifetime of $N_2O_5$ ($\tau_{calc}(N_2O_5) = (k_{N_2O_5} + \frac{k_{NO_3}}{K_{eq}[NO_2]})^{-1}$) were used for determine whether the situation had satisfied steady state (see details in methods section)."

*19.  Figure S5 and Tables S3, S4: The comparison of equilibrium constants based on different mechanism is an important contribution from this paper.  Suggest a figure that makes a comparison between the parameterizations of Keq, either calculated from parameters as in JPL, or from the ratio of k_forward/k_reverse as in the other parameterizations, to show the variability that is used in the current literature.  Figure S5 is useful for specific in-field data, but only shows variation as a function of time without a clear T dependence.  A separate T dependence would clarify this figure.*

Thank you for suggestion and the figure of dependence of Keq parameterization against T is shown as follows. It is inserted in the supplement as figure S6.

[Figure]

*20. This section could make reference to the review article of Chang et al., who consider the T-dependence of two parameterizations (JPL and IUPAC). Chang, W.L., P.V. Bhave, S.S. Brown, N. Riemer, J. Stutz, and D. Dabdub, Heterogeneous Atmospheric Chemistry, Ambient Measurements, and Model Calculations of $N_2O_5$: A Review. Aerosol Science and Technology, 2011. **45**: p. 655-685.*

Thank you for the suggestion. We added this reference to Text S4 which describes the parameterization of Keq coefficient in different database.

*21. Also, in Figure S5, why is the particular time period chosen out of each campaign? In each case, this is 2 hours of data out of several weeks of measurement.*

The reason we choose these particular time periods is that the precursor concentrations and temperature during this period are close to the average condition of each corresponding campaign.

*22. Line 142-143: Specify how the time to reach a valid steady state is defined. Model mixing ratios equal to some fraction of observed ones, for example?*

As per our reply to comment 9, whether each data point satisfied steady state is validated according to the steady state criterion, defined by the deviation between steady state lifetime of $N_2O_5$ ($\tau_{ss}(N_2O_5) = \frac{[N_2O_5]}{k_{R1}[NO_2][O_3]}$) and calculated lifetime of

$N_2O_5$ ($\tau_{calc}(N_2O_5) = (k_{N_2O_5} + \frac{k_{NO_3}}{K_{eq}[NO_2]})^{-1}$). If the deviation exceeds 10% for a data point, it will be considered as out of steady state. Therefore, we define the time for a simulation of a particular case, starting from initialized conditions to meet the steady state criterion described above, as the time to reach a valid steady state.

We specified the definition in text S5 as follows.

"In Figure S8 and Figure S9, a series of sensitivity test provide an assessment of the time a valid steady state needs under several conditions. The most sensitive variables

to the time to reach a valid steady state are $kN_2O_5$, $kNO_3$ and T, the enhancement of which reduces the induction time, facilitating the approach to valid steady state of $NO_3$-$N_2O_5$ system. The time for a simulation of a particular case, starting from initialized conditions to meet the steady state criterion (detailed in the methods section), is defined as the time to reach a valid steady state."

*23. Figure S7: Same comment as above – the top and bottom axes apparently have no correspondence to each other, which is confusing.*

Similar to our reply to the comment on Figure 2, we add clarification to the figure caption here.

*24. Line 182-187 and Figure S9: What is the mechanism that leads to incorrect determination of gamma_N2O5 at increasing k_NO3? Perhaps a plot to demonstrate why the slope or intercept changes by such a large amount would be helpful.*

The increasing $kNO_3$ could amplify the bias of $\gamma(N_2O_5)$ derivation by linear fit even with $kNO_3$ RSD keep the same. When we conduct linear fit by Eq. (5), if the values of $kNO_3$ for each data point vary with the time period used for linear fit, the derived $\gamma(N_2O_5)$, as the intercept, will deviate from the expected value due to this variation.

For example, a positive covariance between $kNO_3$ and $\left(0.25 c S_a K_{eq}[NO_2]\right)^{-1}$ could lead to negative bias of $\gamma(N_2O_5)$ derived therefrom. However, it is hard to visualized this mathematical relationship in the perspective of steady state approximation in this work and we will investigate this subject in future works.

*25. The right axis in Figure S9a also does not match the tick marks on the right axis in panel b, which shows the values on this axis.*

We adjust the tick formats to match each other as follows.

[Figure]